# Evaluation of Prediction Models for the Capping and Breaking Force of Tablets Using Machine Learning Tools in Wet Granulation Commercial-Scale Pharmaceutical Manufacturing

**DOI:** 10.3390/ph18010023

**Published:** 2024-12-27

**Authors:** Sun Ho Kim, Su Hyeon Han, Dong-Wan Seo, Myung Joo Kang

**Affiliations:** 1College of Pharmacy, Dankook University, 119, Dandae-ro, Dongnam-gu, Cheonan-si 31116, Republic of Korea; dwseomb@dankook.ac.kr (D.-W.S.); kangmj@dankook.ac.kr (M.J.K.); 2Department of Mechanical Engineering, Kongju National University, 1223-24, Cheonan-daero, Seobuk-gu, Cheonan-si 31080, Republic of Korea; tngus8581@gmail.com

**Keywords:** machine learning, capping prediction, tablet breaking force, permutation feature importance analysis

## Abstract

**Background/Objectives**: This study aimed to establish a predictive model for critical quality attributes (CQAs) related to tablet integrity, including tablet breaking force (TBF), friability, and capping occurrence, using machine learning-based models and nondestructive experimental data. **Methods**: The machine learning-based models were trained on data to predict the CQAs of metformin HCl (MF)-containing tablets using a commercial-scale wet granulation process, and five models were each compared for regression and classification. We identified eight input variables associated with the process and material parameters that control the tableting outcome using feature importance analysis. **Results**: Among the models, the Gaussian Process regression model provided the most successful results, with *R*^2^ values of 0.959 and 0.949 for TBF and friability, respectively. Capping occurrence was accurately predicted by all models, with the Boosted Trees model achieving a 97.80% accuracy. Feature importance analysis revealed that the compression force and magnesium stearate fraction were the most influential parameters in CQA prediction and are input variables that could be used in CQA prediction. **Conclusions**: These findings indicate that TBF, friability, and capping occurrence were successfully modeled using machine learning with a large dataset by constructing regression and classification models. Applying these models before tablet manufacturing can enhance product quality during wet granulation scale-up, particularly by preventing capping during the manufacturing process without damaging the tablets.

## 1. Introduction

Tablet breaking force (TBF) and friability are measured and controlled to ensure that the tablets are intact during downstream processes such as coating or packing, and are relevant for ensuring dosage-form robustness during manufacturing [1,2]. They are primarily influenced by the bonding mechanisms and the development of contact areas where attractive forces between particles are significant [3]. Inadequate compression pressure can result in weak bonding, which can potentially cause premature disintegration [4]. Additionally, for highly viscoelastic powder materials, high-strain compression may occasionally lead to cracking, which can result in capping [5]. Consequently, patients may be at risk of not receiving one of the components or receiving an incorrect dosage [4]. However, the traditional quality control method is time consuming and requires significant financial and human resources. Furthermore, destructive tests after sampling only detect capping occurrences at the end of manufacturing, which poses a risk of discarding the finished product [6]. Moreover, the complexity of tablet processes, with variations in manufacturing processes, makes TBF, friability and capping occurrence prediction potentially unachievable and unreliable using linear models such as partial least squares [7].

Machine learning tools have been applied in pharmaceutical manufacturing to model and optimize processes, with notable implementations including automated adaptive systems for real-time monitoring, feature selection, and process scale-up, demonstrating significant advantages over traditional multivariate statistical methods [1]. In the emulsion manufacturing process, an automated machine vision and machine learning approach was developed to classify emulsions into quality categories, demonstrating 40% greater accuracy and 180 times faster processing compared to manual assessment [8]. Predictive machine learning models were also applied in the manufacturing of pharmaceutical oral solid dosage forms using historical data from a commercial immediate-release tablet product to predict particle size distribution and tensile strength, enhancing product-related know-how and supporting quality by design approaches [2]. Akseli et al. [1] experimentally evaluated the disintegration time and tablet strength of various formulations, and the results were compared with the predicted values determined by different machine learning-based models.

To date, most critical quality attribute (CQA) prediction studies, including capping defects, have been conducted using one or two components made from common pharmaceutical excipients based on data obtained from image analysis or web libraries. However, the compression of one or two components does not accurately represent real pharmaceutical formulations, particularly granules fabricated using wet granulation methods. Furthermore, capping defect prediction techniques using image analysis are not feasible in pharmaceutical manufacturing because of the high cost, bulky equipment, and extended scan durations required for analysis [9]. To the best of our knowledge, prediction studies on TBF, friability, and tablet capping defects using large-scale nondestructive experimental datasets have not yet been reported to date owing to limited experimental data.

Therefore, this study aimed to evaluate a prediction model for CQAs related to tablet integrity, such as TBF, friability, and capping occurrence with a wet granulation process using machine learning-based models. Then, we determined the input variables associated with the material and process parameters that control the tableting output using permutation feature importance analysis. Metformin HCl (MF) tablets, prescribed to treat type-2 diabetes mellitus, were used as the model products [10], and eight process variables were used as the model inputs to evaluate their impact on the CQA. The specific compositions of the mixtures subjected to granulation are listed in Table 1. In this study, tableting outcomes (TBF, friability, and capping occurrence) were modeled in terms of both qualitative and quantitative responses using machine learning with a large dataset by constructing regression and classification models. The models were trained on the data, and machine learning techniques, including Gaussian Process Regression (GPR), Decision Trees (DT), Random Forests (RF), Boosted Trees (BT), Support Vector Machines (SVM), and k-Nearest Neighbors (kNN), were compared. Subsequently, several process parameters were evaluated to investigate the process and characterize their impact on CQAs.

## 2. Results and Discussion

### 2.1. Data Preparation

For CQA prediction, the following process parameters and material properties were used as input variables: (1) magnesium stearate (MgSt) fraction (%); (2) tablet compression speed (RPM); (3) compression force (N); (4) ejection force (N); (5) weight (mg); (6) thickness (mm); (7) diameter (mm); and (8) tablet porosity (%). It is worth noting that the inclusion of TBF, friability, and capping can be determined via nondestructive tests using easy-to-monitor input variables.

A large dataset of experimental results for tablets was used to construct models with 1428; 762; and 1599 data points for TBF, friability, and capping occurrence, respectively. The model performance was assessed through the CQA prediction. The holdout method was used to ensure the model’s validity. This approach led to the random selection of 15% of the samples for hyperparameter optimization, 15% for testing, to evaluate the model performance, and the remaining 70% for training. Min–max normalization techniques were applied to the raw data to mitigate scaling and weighting issues within the training model. Using the min-max method, all inputs and outputs were normalized to a range of [0, 1].

### 2.2. TBF Prediction

The RF, GPR, SVM, BT, and DT models were applied to the tablet data to model the relationships between input variables and TBF. The model was validated by using training, validation, and test datasets to evaluate the prediction accuracy. The statistical characteristics (coefficient of determination (*R*^2^), root mean square error (RMSE), and mean absolute error (MAE)) of the TBF predictions are summarized in Table 2. A higher *R*^2^ value, along with lower RMSE and MAE values, indicates a better ability of the model to predict the response values accurately [11]. All models for TBF showed outstanding performances, with *R*^2^, RMSE, and MAE values for the RF, GPR, SVM, BT, and DT models of 0.959, 0.959, 0.958, 0.959, and 0.954, 0.780, 0.890, 0.798, 0.787, and 0.863, and 0.561, 0.541, 0.530, 0.595, and 0.609, respectively.

The correlation between the experimental and predicted values for the analyzed parameters is shown in Figure 1. Overall, all five models provided accurate predictions with high *R*^2^ values and low MAE and RMSE values.

### 2.3. Feature Importance Analysis of the Factors Affecting the TBF

When dealing with a large dataset, it is important to establish relationships among the CQAs and input variables, followed by the identification of the significant parameters that correlate with the response [12]. We employed the permutation importance (PI) method to assess the relative importance of each input variable in the predictive models. To assess the model’s functional importance, we calculated the PI by shuffling each input variable individually and measuring the change in performance (RMSE) for each permutation. This approach allowed us to determine the influence of each variable based on its impact on model performance.

Figure 2 shows the relative importance of each input variable in the RF model, which was determined by making predictions on randomly selected data from the training set. The PI method was used to evaluate the significance of input variables. This study involved performing random shuffles of the input variables to evaluate the decrease in model performance via hold-out validation. A longer bar indicates a higher parameter importance. The PI values were 0.297, 0.001, 0.301, 0.010, 0.015, 0.147, 0.022, and 0.195 for the MgSt fraction, tablet compression speed, compression force, ejection force, weight, thickness, diameter, and tablet porosity, respectively. According to the calculations performed using the RF model, the most important individual factor among the input variables was the compression force. Previous studies have demonstrated that the compression force is directly proportional to the TBF of compressed tablets [13,14]. As the granules were compressed, the relative density of the tablet increased to a certain extent, causing a higher TBF compared to its uncompressed state. Additionally, the MgSt fraction ranks higher than the other input variables. Higher MgSt fractions can prevent tight particle-particle binding, thereby disrupting the compressibility of the granule powder and reducing the mechanical strength of the tablet [15]. This suggests the significance of using MgSt as an indicator for the estimation. The other important process parameters indicated by the model are the tablet porosity and thickness, which aligns with common knowledge and the literature [16]. Tablet porosity and thickness have a similar impact on the TBF, which indicates a dependency between the two factors. Meanwhile, tablet compression speed ranks lower than the other factors, which is surprising considering previous knowledge. These findings suggest that the compression of MF (over 80% of the total tablet fraction), which is considered a brittle deforming material, is less speed dependent [17]. This is because the fragmentation of brittle materials is achieved rapidly, and extended exposure to force has a more limited impact on tablet properties.

### 2.4. Friability Prediction

Using a method similar to that used for the TBF, five models were employed for friability prediction. However, each model yielded a different *R*^2^ value. Three models, RF, GPR, and BT, presented satisfactory predictions, with *R*^2^ values of 0.919, 0.949, and 0.917, respectively. In contrast, for the SVM and DT models, *R*^2^ did not exceed 0.850 for the test set. Notably, the use of ensemble methods combined with bagging and boosting significantly improved the accuracy of the pure DT model, with the *R*^2^ value increasing from 0.822 for the pure DT model to 0.919 and 0.917 for RF and BT models, respectively. BT and RF, the single classifiers based on the ensemble learning of DT, find subtle relationships within the data and evaluate them quantitatively, additionally covering more features than single trees because these algorithms builds multiple DTs and combines their outputs, which is consistent with our results [18].

A comparison between the predicted and experimental values is shown in Figure 3. For the RT, GPR, and BT models, a good estimation was achieved based on sufficiently high *R*^2^ values and low MAE values in the validation set. However, the SVM and DT models produced inferior friability predictions compared to the RF and BT models. These results indicate that DT combined with optimization techniques, such as boosting and bagging methods, can avoid issues related to high variance and is better suited for modeling relationships between tablet parameters and friability. Nevertheless, the prediction accuracy and error for friability were slightly inferior to those of the TBF (Table 2 and Figure 3), suggesting that some friability-related processes or material parameters included in the input are inadequate. For example, granulation moisture content is an important property that has a strong impact on friability [19] but was not included in our material data. In future work, such new properties should be added to further reduce the prediction errors.

### 2.5. Feature Importance Analysis of the Factors Affecting Friability

The feature importance of friability is presented in Figure 4. In the RF model, the major factor was compression force, with a PI value of 0.734, showing a pattern similar to that of the TBF. The thickness of the intact tablet was the second most significant variable, with a PI value of 0.113. The relationship between thickness and inter-particulate bonding strength of pharmaceutical tablets has been previously reported [20]. The compressed volume of tablets with lower thickness has a positive effect on friability by offering a higher number of molecular contacts and enhancing tablet strength, which is consistent with our results.

### 2.6. Tablet Capping Occurrence Prediction

The tablet capping prediction was evaluated in terms of categorical responses using the RF, DT, SVM, BT, and kNN models. The final model, developed using the hold-out method, yielded the true positive (TP), true negative (TN), false positive (FP), false negative (FN) accuracy, and percent accuracy for capping occurrence in both the training and test sets (Table 3).

### 2.7. Feature Importance Analysis for the Factors Affecting Capping Occurrence

The relative importance of each variable to capping occurrence based on the RF model is shown in Figure 5. The PI value of each factor affecting the capping occurrence followed a pattern similar to that of the TBF and friability models. Among the parameters, compression force was the most important, and an increase in the MgSt fraction led to a greater capping probability. The importance of the lubricant is not surprising, despite its small amount in the formulation, given that MgSt among the granules often deteriorates tablet integrity and breaks the granules, leading to an impact on capping. Therefore, capping can be predicted based on the increase in compression force and MgSt fraction. The tablet compression speed, ejection force, weight, thickness, diameter, and porosity were not related to capping occurrence.

### 2.8. Strength and Limitations

The major strength of this study is the large scale of the data, and all of the data are based on experimental results, which provide confidence in the model. A commercial-scale batch (300,000 tablets) was fabricated using wet granulation methods, including the API and several excipients, to represent real pharmaceutical formulations. A further strength of this study is the use of nondestructive input variables for predicting product quality, indicating feasible methods for cost savings in the pharmaceutical industry.

The main limitation of this study is that it does not demonstrate the applicability of the models to other tablet formulations, such as chewable or sublingual tablets. Furthermore, compatibility with dry granulation and direct compression manufacturing methods for tablets was not covered in this study.

In future work, other formulations, such as capsule or granule formulations, should be included to evaluate the model’s applicability. It may be difficult to apply our models to other formulations, which would require new models to be trained and constructed for predicting CQAs. However, as other similar types of tablets to those to this study, such as bilayer tablets and tablets with various shapes, have input variables and CQAs that do not differ significantly from this study, they are expected to be compatible with our models with minimal parameter tuning.

## 3. Materials and Methods

### 3.1. Materials

MF (purity over 99 *w*/*w*%; median diameter: 38.33 μm) was purchased from Granules India Limited (Madhapur, Hyderabad, India), as described in [21]. Polyvinylpyrrolidone (PVP K30) was obtained from BASF (Ludwigshafen Land, Rheinland-Pfalz, Germany). Methacrylic acid copolymers (Eudragit S100) and high-viscosity-grade HPMC2208 were acquired from Evonik (Essen, NRW, Germany) and Dow Chemicals (Montgomeryville, PA, USA), respectively. Carbomer 934P (Carbopol^®^ 934PNF) was obtained from BF-Goodrich (Cleveland, OH, USA). MgSt was obtained from FACI Asia Pacific (Merlimau Pl., Jurong Island, Singapore).

### 3.2. MF-Loaded Granule Using the Wet Granulation Method and Tablet Preparation

The MF-loaded granules were fabricated through wet granulation into a batch of 300,000 tablets as described in [21]. The specific composition of the mixtures used in the granulation process was prepared with reference to the previous study [21] and is listed in Table 1. MF-loaded granules were manufactured by dissolving the binders in an ethanol–water solution, spraying the mixture onto the drug powder, blending with excipients, and followed by drying and sieving the mixture. MgSt, with a fraction of 0.6–2.0% of the total mixture, sieved using a 40-mesh sieve, was added to the mixture and lubricated at 10 rpm for 5 min.

To prepare the tablets, the MF-loaded granules were filled into a die. The granules were then compressed using a tablet press machine (TDP5, LFA, Taichung, Taiwan). The tablet press machine was equipped with a commercial Euro standard D441 punch and a 9.8 mm-diameter round convex die. The tablet compression and ejection force with speeds set ranging from 60 to 80 rpm were determined using 10,000 lb capacity rod cell (LCM375, FUTEK, Irvine, CA, USA) with 0.02 mV of minimum resolution in the lower punch. The sensor continuously measured and recorded the force and distance until the tablet completely exited the die. The peak forces derived from the force–displacement plots were determined as the compression and ejection forces.

The tablet thickness and diameter were measured using a digital micrometer (Mitutoyo 395–251, Mitutoyo, Tokyo, Japan) immediately after the tablets were ejected. The capping occurrence of tablets was macroscopically investigated by assessing the presence of capping phenomena during the friability and TBF test.

### 3.3. Porosity of the MF-Loaded Tablets

To determine the compressed tablet porosity (*ε*), the true density was determined by measuring the true volume of the sample using a helium Ultrapyc 1220e Automatic Gas Pycnometer (Anton Paar QuantaTec Inc., Boynton Beach, FL, USA), as described in [22]. *ε* was calculated as follows:(1)$$\varepsilon =\left(1-\frac{m}{\rho_{t}v}\right)\times 100$$
where *ρ_t_* is the true density, *m* is the tablet weight, and *v* is the tablet volume. The apparent volume of tablets was determined using a 3D modeling program (CATIA V5R21, Dassault Systemes, Vélizy-Villacoublay, France).

### 3.4. Compaction Breaking Force Required to Break the MF-Loaded Tablets

The compaction force required to break the MF-loaded tablets was assessed using a tablet combination tester (Multicheck VI, Erweka, Heusenstamm, Germany), as described in [16]. The prepared tablets were placed between two plates and subjected to compression directly onto the tablet surface, resulting in tablet fracture. The peak force derived from the force–displacement plots was determined as the compaction force. Although the term ’hardness’ is widely used and recognized in pharmacopeias, the more specific term ’breaking force’, as noted in the USP, was used in this study [23].

### 3.5. Friability Test

Friability tests were conducted on the prepared tablets using a friability tester (PT F20E; Pharma Test, Hainburg, Germany). The tablets were then placed in a drum and rotated 100 times at 25 rpm. The mass loss was measured by weighing the tablets before and after testing.

### 3.6. Machine Learning Techniques

Data were analyzed using MATLAB^®^ 2024a (MathWorks, Natick, MA, USA). The input variables for the models included the mechanical parameters of the tablets, as well as material parameters such as the lubricant amount, weight, diameter, thickness, tablet porosity, tablet compression speed, compression force, and tablet ejection force, which may have direct or indirect effects on the TBF, friability, and capping occurrence. As this study aimed to develop a practical and feasible model that can be applied to more specific applications, we intentionally excluded factors that are formulation-related and thus difficult to change owing to regulatory issues, such as active substances and binder fraction. Furthermore, porosity was excluded as an input variable from the friability prediction model according to expert judgment.

In the prediction, several models were explored by fitting and evaluating them on the data, preprocessing with the min–max normalization technique, and studying the leave-one-out cross-validation errors. Machine learning techniques, including RF, GPR, SVM, BT, DT, and kNN, were used to predict the TBF, friability, and capping occurrence. Each model has advantages that can be effectively applied to the dataset.

DT is a classification method that partitions data into multiple subsets, identifies the splitting criteria most relevant to the response in a subset, and recursively applies these criteria to track relationships [12]. DT was expected to be suitable for understanding the relationship between input variables and CQAs. In the study, DT was used to identify the impact of input variables such as lubricant content and compression pressure on quality attributes. With its simple and rapid learning using a single model, it was effectively utilized in the initial data analysis stage.

BT combines multiple trees through an ensemble approach to reduce the variance of the dataset and improve prediction performance [24]. The model is useful for dealing with data that include a low rate of defective data, with stable prediction performance. BT was expected to be suitable for the datasets in this study, which include a low proportion of defective data, for evaluating and classifying defects, as well as predicting CQAs. Additionally, BT was employed to precisely learn the complicated relationship between compression pressure and ejection pressure, and it was expected to achieve high accuracy.

RF, initially introduced by Breiman [25], is an ensemble approach that combines a set of classification or regression trees into a single model, selecting the predicted response based on a majority vote or average value from individual trees [2]. RF was expected to provide superior performance in predicting TBF and friability for the data in this study by effectively modeling the nonlinear and interactive relationships between process variables such as compression pressure, ejection pressure, and lubricant content. Furthermore, RF was used for the quantitative analysis of feature importance and the impact of process variables on CQAs.

SVM determines the separating hyperplane by transforming samples into a high-dimensional Hilbert space, maximizing the distance to the nearest training samples [1]. SVM was employed to effectively learn from the data in this study, which included complex relationships and nonlinear boundaries, by using the Radial Basis Function (RBF) kernel. Specifically, SVM was expected to provide superior performance in separating normal and defective data within a high-dimensional space in the classification model.

GPR was expected to be suitable for the data in this study, which include subtle variables that can influence CQAs such as hardness and friability. Moreover, GPR was anticipated to quantify the uncertainty of results using prediction intervals, thereby enhancing the model reliability.

To predict tablet defects, the qualitative response was assigned as a capping occurrence (Yes or No) for each tablet in the dataset. Classification, which is a method for predicting categorical responses for capping occurrence prediction, includes DT, RF, SVM, BT, and kNN. Among these, kNN can commonly be employed for classification and regression problems. t-SNE is conceptually connected with kNN in that it relies on distance-based similarity and local data relationships. By reducing high-dimensional data to lower dimensions, it is a highly useful technique for visualizing data structures and clusters with high accuracy, which has made it widely used in many studies [26,27]. Nevertheless, we expected that kNN, using a supervised learning-based classification algorithm, would be more suitable for classifying defective tablets in our study. kNN is effective at utilizing labeled data to perform classification tasks, which fits with the process and data characteristics in this study, such as the nonlinear relationships between input variables like lubricant contents, compression pressure, and ejection pressure.

### 3.7. Evaluation of the Results

A dataset comprising 1599 data points, constructed using hold-out validation, was split into training, validation, and test subsets [28]. The training set (70% of all cases) was used to train the model, and the validation set (15%) was used exclusively for hyperparameter tuning in all models. After model selection, the test set (the remaining 15%) was reserved to evaluate the final predictive performance of the unknown data. Stratified random sampling was applied to ensure a balanced representation of each class in the training, validation, and test sets. This process was repeated across 15 different subsets to evaluate the prediction variability and feature importance based on the selected sample combinations.

Three statistical metrics—*R*^2^, MAE, and RMSE—were employed to evaluate the model performance. These metrics were calculated as follows:


(2)
$$\mathrm{RMSE}=\sqrt{\frac{1}{n}\sum_{i=1}^{N}{\left(y_{i}-y_{i}\right)}^{2}}$$


and


(3)
$$R^{2}=1-\frac{\sum_{n}{\left(y_{i}-{\hat{y}}_{i}\right)}^{2}}{\sum_{n}{\left(y_{i}-{\bar{y}}_{i}\right)}^{2}}$$


and

(4)$$\mathrm{MAE}=\frac{1}{n}\sum_{i=1}^{n}\left|{\hat{y}}_{i}-y_{i}\right|$$
where *y*_i_ and ${\hat{y}}_{i}$ represent the experimental and predicted values, respectively; ${\bar{y}}_{i}$ is the mean value of the training set; and *n* represents the number of samples.

The classification model developed from the training dataset was used to predict the capping occurrence in the test data. Model performance was evaluated based on the proportion of correctly predicted instances in the test set. The accuracy or misclassification rate was calculated, with the terms for model accuracy and misclassification defined as follows [12]:(5)$$Classification accuracy=\frac{TP+TN}{Total}$$

and
(6)$$Misclassification Rate=\frac{FP+FN}{Total}$$
where true positive (TP) represents tablets predicted as “Yes” that are actually capped, true negative (TN) represents tablets predicted as “No” that are not capped, false positive (FP) represents tablets predicted as “Yes” that are not actually capped, and false negative (FN) represents tablets predicted as “No” that are actually capped.

### 3.8. Relative Importance of Each Input Variable

To evaluate the relative importance of each input variable, the permutation importance method was applied for each feature. This approach involves individually shuffling the values of each feature in a trained model to measure its effect on the response, specifically by observing the relative increase in RMSE [29]. When the value of a feature is permuted, the resulting performance decrease (increase in RMSE) reflects the significance of that feature in maintaining the prediction accuracy [30]. Each feature importance score was averaged across 100 permutations to ensure robustness. Probability normalization techniques were applied to mitigate scaling issues within the model.

## 4. Conclusions

In this study, we successfully applied machine learning-based models to predict CQAs such as TBF, friability, and capping in MF tablets produced by commercial-scale wet granulation using nondestructive experimental data. All models we suggested for the TBF exhibited outstanding performance (*R*^2^_RF_ = 0.959; *R*^2^_GPR_ = 0.959; *R*^2^_SVM_ = 0.958; *R*^2^_BT_ = 0.959; and *R*^2^_DT_ = 0.954). For friability prediction, the most successful models were the GPR and RF models, with *R*^2^ values of 0.949 and 0.919, respectively, using regression, regularization, and ensemble techniques. Furthermore, the importance of the compression force and MgSt fraction as input variables was demonstrated by the PI score of the regression model. The potential of machine learning-based models to accurately predict capping occurrence was further validated, achieved over 95% accuracy across all models. These findings indicate that advanced data-driven approaches, utilizing nondestructive and easy-to-monitor tools, can be effectively applied in pharmaceutical manufacturing to enhance product quality. In particular, these methods can predict tablet defects, such as preventing capping before completing the manufacturing process in wet granulation scale-up, thereby improving the efficiency and reducing waste.

## Figures and Tables

**Figure 1 pharmaceuticals-18-00023-f001:**
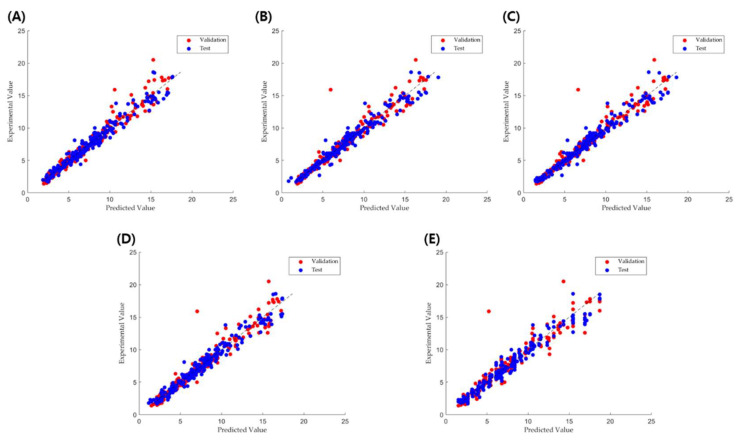
Experimental vs. predicted correlation for the (**A**) RF, (**B**) GPR, (**C**) SVM, (**D**) BT, and (**E**) DT models for TBF prediction.

**Figure 2 pharmaceuticals-18-00023-f002:**
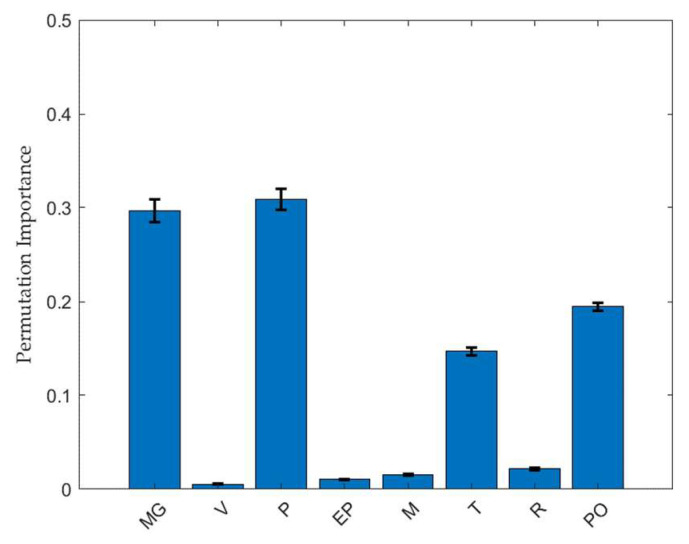
Permutation feature importance results for the TBF based on the RF model. Data are represented mean ± standard deviation (*n* = 10). Abbreviations: MG, magnesium stearate; V, tablet compression speed; P, compression force; EP, ejection force; M, weight; T, thickness; R, diameter; PO, porosity.

**Figure 3 pharmaceuticals-18-00023-f003:**
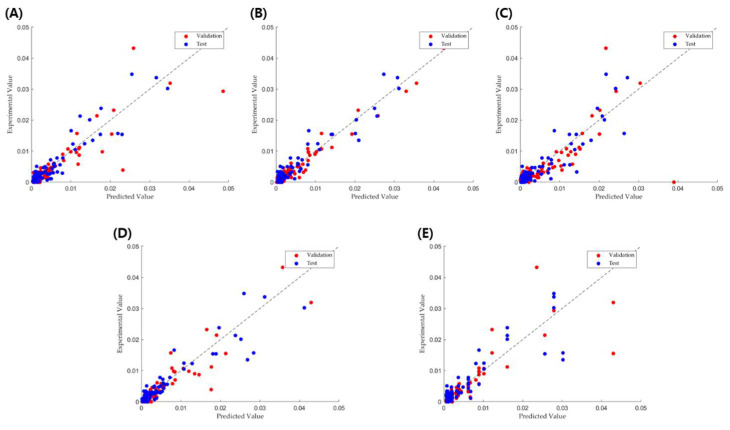
Experimental vs. predicted correlation for the (**A**) RF, (**B**) GPR, (**C**) SVM, (**D**) BT, and (**E**) DT models for friability prediction.

**Figure 4 pharmaceuticals-18-00023-f004:**
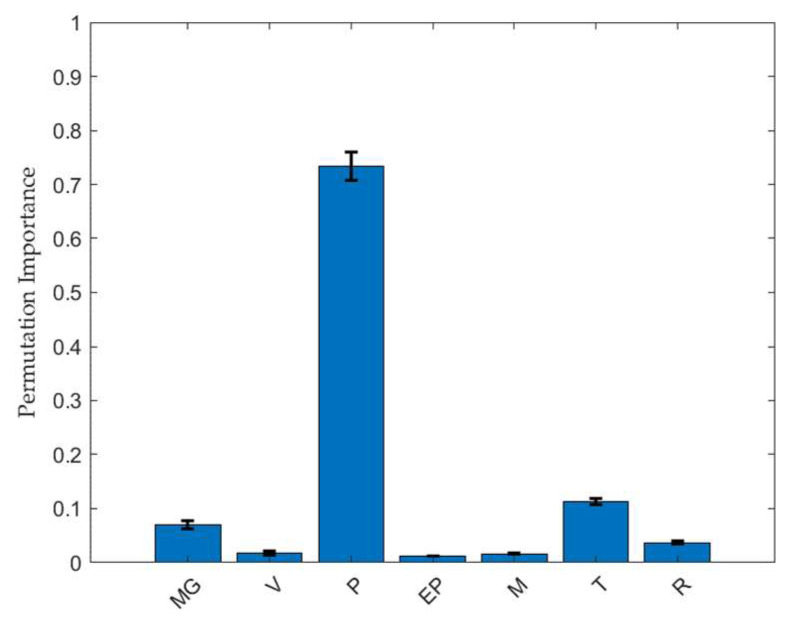
Permutation feature importance results for friability based on the RF model. Data are represented mean ± standard deviation (*n* = 10). Abbreviations: MG, magnesium stearate; V, tablet compression speed; P, compression force; EP, ejection force; M, weight; T, thickness; R, diameter.

**Figure 5 pharmaceuticals-18-00023-f005:**
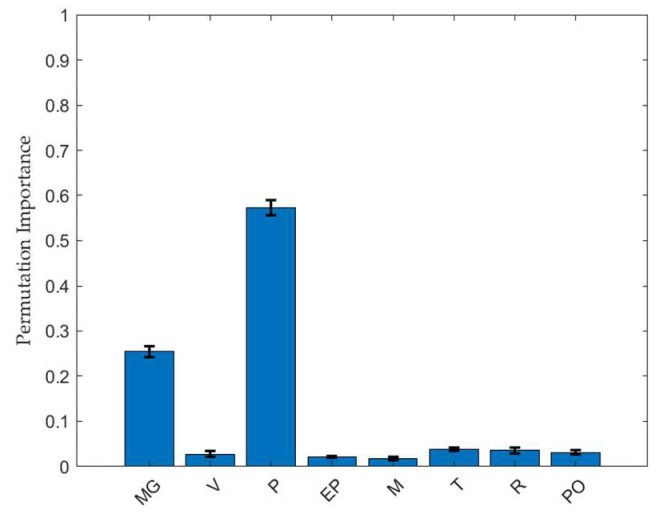
Permutation feature importance results for capping occurrence based on the RF model. Data are represented as mean ± standard deviation (*n* = 10). Abbreviations: MG, magnesium stearate; V, tablet compression speed; P, compression force; EP, ejection force; M, weight; T, thickness; R, diameter; PO, tablet porosity.

**Table 1 pharmaceuticals-18-00023-t001:** Composition of mixtures subjected to granulation processes.

Function	Ingredient	
Active pharmaceutical ingredient	MF	500
Binder	Polyvinylpyrrolidone	20
Lubricant	Mg stearate	3.6–12.3
Controlled release excipient	Carbomer 934P	10
Controlled release excipient	HPMC2208	50
Controlled release excipient	Methacrylic acid copolymer	20

Note: The MgSt fraction was formulated as 0.6% to 2.0% of the total mixture.

**Table 2 pharmaceuticals-18-00023-t002:** *R*^2^, RMSE, and MAE comparison for TBF and friability prediction for different machine learning-based models using input variables.

Objective Variable	Model	*R* ^2^	RMSE	MAE
TBF	RF	0.959	0.780	0.561
	GPR	0.959	0.890	0.541
	SVM	0.958	0.798	0.530
	BT	0.959	0.787	0.595
	DT	0.954	0.863	0.609
Friability	RF	0.919	0.002	0.001
	GPR	0.949	0.002	0.001
	SVM	0.764	0.004	0.002
	BT	0.917	0.003	0.001
	DT	0.822	0.004	0.002

**Table 3 pharmaceuticals-18-00023-t003:** Prediction accuracy for capping occurrence for different machine learning-based models using input variables.

	Model				
Indicator	RF	DT	SVM	BT	kNN
TP	21	21	16	27	20
TN	210	207	217	201	212
FP	3	7	2	7	5
FN	4	3	3	3	1
Accuracy (%)	97.06	95.80	97.90	95.80	97.48
Misclassification (%)	2.94	4.20	2.10	4.20	2.52
No. of samples	238	238	238	238	238

The models can be ranked in terms of performance in the following order: SVM (97.90%), kNN (97.48%), RF (97.06%), BT (95.80%), and DT (95.80%). All models exhibited an accuracy of >95% for capping occurrence, which indicates that they have sufficient prediction accuracy.

## Data Availability

The data supporting the findings of this study are available from the corresponding author upon reasonable request.

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
