# Peer review of "Evaluation of Prediction Models for the Capping and Breaking Force of Tablets Using Machine Learning Tools in Wet Granulation Commercial-Scale Pharmaceutical Manufacturing"

_pharmaceuticals, 2024, doi:10.3390/ph18010023_

Round 1

Reviewer 1 Report

Comments and Suggestions for Authors

See attachment

Reviewer 2 Report

Comments and Suggestions for Authors

The article titled "Evaluation of Prediction Models for Capping and Breaking Force of Tablets Using Machine Learning Tools in Wet Granulation Commercial-Scale Pharmaceutical Manufacturing" presents an innovative approach to integrating machine learning into pharmaceutical manufacturing, specifically targeting the prediction of critical quality attributes in tablet production. This work contributes significantly to the optimization of large-scale wet granulation processes and ensures product quality with enhanced precision. The study targets a significant problem in commercial-scale pharmaceutical manufacturing, specifically optimizing CQAs such as tablet breaking force, friability, and capping occurrence. The comments are as follows that need to be addressed.

1.       Tablet breaking force? Hardness of the table is not a better word as it is widely used and official in pharmacopoeias.

2.       Introduction needs to improve the addition of machine learning tools/study in pharma manufacturing scale Pharmaceutical. There is no need for patient safety and effective use examples (it can be removed).

3.       Table 1. Composition of mixtures subjected to granulation processes. On what basis was this composition selected?

4. Figure, 2,4 and 5 should have error bars.  

5.       While the results are promising, elaborate on why specific ML models were chosen for regression and classification.

6.       Highlight or discuss any limitations, such as the applicability of the models to different tablet formulations or manufacturing conditions, to provide a balanced perspective.

7. Discuss the applicability of these models to other formulations or manufacturing processes in the healthcare sector.

8.       Page 10: The equations should be numbered.

Round 2

Reviewer 1 Report

Comments and Suggestions for Authors

The authors have addressed my minor review comments; I support publication